# The Effects of Physiotherapy in the Treatment of Cubital Tunnel Syndrome: A Systematic Review

**DOI:** 10.3390/jcm11144247

**Published:** 2022-07-21

**Authors:** Tomasz Wolny, César Fernández-de-las Peñas, Tomasz Buczek, Magdalena Domin, Arkadiusz Granek, Paweł Linek

**Affiliations:** 1Institute of Physiotherapy and Health Sciences, Musculoskeletal Elastography and Ultrasonography Laboratory, The Jerzy Kukuczka Academy of Physical Education, Mikołowska 72A, 40-065 Katowice, Poland; magdalena.rutka@wp.pl (M.D.); p.linek@awf.katowice.pl (P.L.); 2Department of Physical Therapy, Occupational Therapy, Rehabilitation and Physical Medicine, Universidad Rey Juan Carlos, 28922 Madrid, Spain; cesar.fernandez@urjc.es; 3Physiotherapy Clinic, “FizjoMedical”, 44-251 Rybnik, Poland; tomaszbuczek91@gmail.com; 4Hospital of the Ministry of Interior and Administration, 25-316 Kielce, Poland; arkadiusz.granek@gmail.com

**Keywords:** cubital tunnel syndrome, ulnar neuropathy, physiotherapy modalities, treatment outcomes, review

## Abstract

Background: To date, various forms of physiotherapy are used in the treatment of cubital tunnel syndrome (CuTS). The effectiveness of physiotherapy for CuTS is inconclusive. The aim of this systematic review was to evaluate the effects of physiotherapy in the conservative treatment of CuTS. Methods: The six databases were searched from December 2020 to March 2022. The inclusion criteria were randomised controlled trials, case series, and case reports that evaluate the effects of physiotherapy in the treatment of adult participants with diagnosis CuTS. A total of 11 studies met the eligibility criteria, capturing a total of 187 participants. Results: In three types of papers, pain, muscle strength, and limitation of upper limb function were the most frequently assessed characteristics. Physiotherapy was most often based on manual therapy, neurodynamic techniques, and electrical modalities. One clinical trial rated the risk of bias “high” and the other two “some concerns”. In case-series designs, five studies rated the risk of bias as “serious” and three as “moderate”. Most of the studies showed a significant improvement in the clinical condition, also in the follow-up study. Only one clinical trial showed no therapeutic effect. Conclusion: There is no possibility of recommending the best method of physiotherapy in clinical practice for people with CuTS based on the results of this systematic review. More high-quality studies are required.

## 1. Introduction

Cubital tunnel syndrome (CuTS) is a compressive neuropathy of the ulnar nerve. It is the second most prevalent peripheral neuropathy of the upper extremity after carpal tunnel syndrome [1,2]. The mean annual incidence of CuTS is estimated at 24.7 cases per 100,000 people [3], and its prevalence is 2–6% in the general population [4]. In the early stages of CuTS, sensory symptoms such as paraesthesia and slight hypoesthesia are reported, occurring mostly paroxysmally and related to the position of the elbow. Over time, these symptoms worsen. This is followed by motor disturbances, mostly weakness and atrophy of the intrinsic muscles of the hand [1,5]. The stages of this neuropathy can be divided into three degrees of severity: mild, moderate, and severe [6]. The progressive course of CuTS over time leads to the impairment of hand function, which adversely affects the activities of daily living, and professional life, and deteriorates the overall health-related quality of life. With the significant prevalence of CuTS, this neuropathy is a major medical, economic, and social problem.

CuTS is referred to as compression neuropathy that occurs around the cubital tunnel. The most common sites of potential compression of the ulnar nerve are the arcade of Struthers, the medial intermuscular septum, the medial epicondyle, the cubital tunnel, and the deep flexor–pronator aponeurosis [7]. This neuropathy is usually divided into primary (idiopathic) and secondary (symptomatic) [8]. In idiopathic forms of CuTS, no morphological abnormalities can be found in the tissues surrounding the ulnar nerve [8]. Furthermore, there are several causes of secondary forms of CuTS (anatomical changes after trauma, degenerative changes, systemic diseases such as rheumatoid arthritis, lipomas, ganglion cysts, inflammatory processes, etc.) [8,9,10]. Other risk factors for CuTS are related to upper extremity motor activity, overhead activity, heavy physical work, obesity, and nicotinism [11]. Therefore, it can be concluded that CuTS is just a synonym for ulnar nerve neuropathy occurring in the elbow area. The difficulty in finding the aetiology of CuTS also affects the choice of the most appropriate treatment modality for this neuropathy.

The treatment of CuTS is divided into surgical and conservative [12]. Palmer and Hughes [10] showed various surgical techniques for ulnar nerve decompression and emphasised that no “gold standard” for surgical treatment has been developed to date. In situ decompression, intramuscular transposition, subcutaneous transposition, submuscular transposition, medial epicondylectomy, and endoscopic techniques are most commonly used [10]. However, surgical treatment is only recommended when muscle strength is weakened, and conservative methods do not bring the expected therapeutic effect [13]. Hence, conservative treatment is used as first-line therapy, usually in the early and mild-to-moderate stages of CuTS [5]. Conservative treatment includes modification of activities of daily living with the avoidance of prolonged elbow joint flexion [10], nonsteroidal anti-inflammatory drugs, steroid injections, and physical therapy approaches [14]. To the best of our knowledge, there is also no standardised procedure for CuTS conservative treatment with well-documented effectiveness. It seems that physiotherapy is one of the most important forms of conservative treatment, which has been proven to be effective in other peripheral neuropathies [15,16,17,18,19]. To date, various forms of physiotherapy have been used in the treatment of CuTS [20,21,22,23]. However, the effectiveness of physiotherapy intervention as a conservative treatment of CuTS is inconclusive. Therefore, it was decided to conduct, for the first time, a systematic review of papers evaluating the effectiveness of physiotherapy treatment for CuTS. Such an analysis determines the quality of the research conducted to date and provides directions for future research.

## 2. Methods

This systematic review adheres to the Preferred Reporting Items for Systematic Reviews and Meta-Analyses (PRISMA) statement [24]. The protocol was registered in the International Prospective Register of Systematic Reviews (PROSPERO) database, registration number CRD 42020219297. 

### 2.1. Data Sources and Searchers

The literature review was conducted between December 2020 and March 2022. Six electronic databases (MEDLINE via PubMed, Cochrane, Embase, Web of Science, Scopus, and PEDro) were searched to identify relevant papers. The search strategy was developed during a panel meeting after an initial article search. It was based on the use of key phrases and/or their abbreviations based on a metadata system (MeSH) and various combinations of these phrases to increase search efficiency. An extensive list of terms to describe the target population based on the PICO acronym was formulated:
P (population)—cubital tunnel syndrome; I (intervention)—physiotherapy treatment; C (comparator)—control group, placebo group, and sham therapy;O (outcomes)—nerve conduction study, discrimination and threshold sensation, functional assessment, and ultrasound imaging measurements (Appendix A). 

All papers were accepted regardless of the year of publication. However, the search was limited to papers available in English. Titles and abstracts of scientific papers retrieved from the databases were analysed for inclusion criteria. The papers that did not show relevance to the subject area studied were excluded. The lists of references in the publications included in the review were also analysed to make sure that other papers that may meet the inclusion criteria were not missed. The resulting papers were combined using the EndNote x9 software (version 19.2.0.13018, Philadelphia, PA, USA).

### 2.2. Study Selection

The review of the retrieved papers was conducted in two stages. The first step was to review the titles and abstracts of papers identified as potentially relevant to the research questions. In the next step, the full texts of the papers identified during the initial selection were reviewed. In both stages, the review was performed by two independent reviewers (T.W., T.B.), and by a third independent reviewer (P.L.) in contentious cases. The inclusion criteria were all experimental studies such as randomised controlled trials (RTCs) and case reports that evaluate the effects of physiotherapy in the treatment of adult participants (>16 years old) with diagnosis CuTS. Participants not diagnosed as CuTS or exposed to any form of surgical procedure and/or with other neuropathies of the upper limb were excluded from the study. The characteristics of the RTC and CS types of studies were presented separately.

### 2.3. Data Extraction

Two reviewers independently extracted and documented data from each included study using Excel (Microsoft, Redmond, WA, USA) according to the Centre for Reviews and Dissemination recommendations [25]. We extracted data including the year of publication, study design, sample size, gender, age, target population, description of interventions, outcome measures, and study results. The main study outcomes expressed as means and SDs were also extracted.

### 2.4. Methodological Quality Assessment

The methodological quality of randomised clinical trials was assessed using the revised Cochrane risk-of-bias tool for randomised trials (RoB 2) [26]. According to the Cochrane guidelines, this tool evaluates possible errors as follows: “low risk”, “some concern”, or “high risk”. Due to the fact that the remaining works that qualified for the review were case studies, their quality was assessed in two stages. In the first stage, the article quality was assessed using The Critical Appraisal Checklist for Case Reports developed by Moola et al. [27]. If five of the eight evaluation criteria are met, the quality is assessed as satisfactory (such a study was included). In the second stage, the quality was assessed using the Risk of Bias in Non-randomised Studies of Interventions (ROBINS-I) assessment tool [28]. This tool evaluates possible errors as follows: “low risk of bias”, “moderate risk of bias”, “serious risk of bias”, “critical risk of bias”, or “no information”. In both cases, the evaluation was conducted by two independent reviewers (T.W. and T.B.). The Cohen κ statistic was applied to determine the agreement between assessors. 

### 2.5. Data Synthesis and Analysis

The extracted data from all included studies were tabulated, including the study authors and sample characteristics, the measurements of the outcome variables, and key results. All the identified studies were included in a qualitative synthesis and are presented in the tables. Initially, it was intended to synthesise the data quantitatively by conducting a meta-analysis. However, because of high heterogeneity in terms of study design, population examined, and various interventions, we could not perform a meta-analysis. 

## 3. Results

### 3.1. Study Selection

An initial search of the databases together with a manual search and analysis of the references identified 1719 papers. This number was reduced to 995 after deleting duplicates. Based on exclusion and qualification criteria, 16 papers met the eligibility criteria (Figure 1). Of these papers, five were excluded after further analysis because they neither concerned CuTS patients nor addressed physiotherapy management. One paper discussed ulnar tunnel syndrome, another described the case of a patient who developed CuTS due to venous thrombosis, while another dealt with the ultrasonographic diagnosis of CuTS. One paper focused on the surgical treatment of CuTS caused by anconeus epitrochlearis, and one evaluated the effectiveness of electrostimulation after traumatic ulnar nerve injury. Finally, 11 articles were included in the final review comprising 3 RCT-type studies [13,29,30] (Table 1) and 8 case-series-type studies [20,21,22,23,31,32,33,34] (Table 2).

### 3.2. Randomised Controlled Trials

#### 3.2.1. Participants

In all RCTs, the subjects were clinically/neurophysiologically diagnosed with CuTS (two studies clinically and neurophysiologically; one study only clinically). A total of 163 subjects aged 16–79 years old were examined and underwent therapy. The RCT involved 87 women and 76 men [13,29,30].

#### 3.2.2. Outcome Measures

All studies evaluated pain (3/3 papers) and muscle strength (3/3 papers) [13,29,30]. Two papers evaluated upper extremity function and nerve conduction [13,30]. In addition, one study assessed sensory threshold [30] and overall health quality [29].

#### 3.2.3. Interventions

Each study used a different therapeutic procedure: orthosis (experimental intervention), neurodynamic techniques (experimental intervention), ergonomic physical activity instruction (control group) [13], continuous shortwave diathermy (experimental intervention) and placebo (control group) [29], and low-level laser therapy (experimental intervention) and ultrasound (experimental intervention) [30].

#### 3.2.4. Risk of Bias

Based on the reviewers’ assessment using the ROB 2 tool to evaluate the risk of bias, the overall bias was considered to be “high” in one study [13], while “some concerns” were indicated in the remaining two RCTs [29,30] (Table 3). The most common flaws were the high risk of bias in domain 4 (risk of bias in measurement of the outcome) in one RTC [13] and domain 2 (risk of bias due to deviations from the intended interventions) in all RTC studies [13,29,30].

#### 3.2.5. Synthesis of the Results

Two RCTs reported significant improvements in the clinical condition of the subjects (pain reduction, improved function, increase in muscle strength, improvement in the sensory threshold, and improvement in nerve conduction), which occurred not only after the therapy but also persisted at 1, 3, and 6 months of follow-up [13,30]. In one study, there were no changes in pain muscle strength, function, and overall health evaluated after therapy in comparison with the control group [29].

### 3.3. Case Studies

In all case series [21,22] and case reports [20,23,31,32,33,34], subjects were clinically and neurophysiologically diagnosed with CuTS (four studies clinically and neurophysiologically; four studies only clinically). In eight case studies, 24 subjects ranging in age from 17 to 71 years were examined and treated. The gender of the subjects was determined in six of the eight papers [20,23,31,32,33,34]. Studies by Oskay et al. and Shen et al. [21,22] failed to specify gender. In total, 48 women and 5 men participated in the study, and in 14 cases, the gender was not specified. 

#### 3.3.1. Outcome Measures

The most frequently evaluated symptom was pain (six of the eight papers) [20,21,22,23,33,34]. Symptom provocation tests (five of the eight papers) [20,21,31,32,34] and functional limitations of the upper extremity (five of the eight papers) were also frequently assessed [21,22,23,32,34]. Muscle strength was assessed in three of the eight papers [21,23,31]. Range of motion was assessed in two papers [20,32] as was the severity of paraesthesia [20,31]. In two of the eight papers, authors evaluated subjective improvements following therapy [23,33]. Furthermore, one study evaluated nerve conduction [33] and sensory threshold [21].

#### 3.3.2. Interventions

Neurodynamic techniques (3/8 papers) [21,32,34] and chiropractic manipulation (two of the eight papers) [20,31] were the most frequently used in therapy. Other papers used dry needling [23], percutaneous electrical stimulation [34], pulsed radiofrequency [33], combined ultrasound and cold therapy [21], and extracorporeal shock wave therapy [22]. 

#### 3.3.3. Therapeutic Effect

In all case studies, significant improvements in the clinical status of the subjects (reduction in pain and subjective symptoms, improved function, increase in grip and pinch strength, reduction in symptoms in provocation tests such as Tinel’s sign or elbow flexion test) were reported, which occurred not only after therapy [20,21,22,23,31,32,33] but were also maintained in the 6-, 8-, 10- and 12-month follow-up periods [21,22,23,32,33,34].

#### 3.3.4. Risk of Bias

The Critical Appraisal Checklist for Case Reports values ranged from 5 to 7 in all case studies, which was considered sufficient [20,21,22,23,31,32,33,34] (Table 4). In the second stage based on the reviewer’s evaluation using the ROBINS-I tool for non-randomised studies of interventions to evaluate the risk of bias, the overall bias was serious in five of the eight papers [20,21,31,32,34] and moderate in three of them [22,23,33] (Table 5). The most common flaws were a serious risk of bias in “bias due to confounding” in five of the eight studies [20,21,31,32,34]; “bias in classification of interventions” in five out of the total eight papers [20,21,31,32,34]; “bias due to deviations from intended interventions” in five of the eight studies [20,21,32,34]; “bias in measurement of outcomes” in another combination of five studies out of the total eight [20,21,31,32,34], and the “overall bias” in five of the eight papers [20,21,31,32,34].

## 4. Discussion

The aim of the present systematic review was to evaluate the effects of physiotherapy in the conservative treatment of CuTS. Based on the search strategy used, 11 papers were included in the review, consisting of 3 RCTs [13,29,30] 2 case series [21,22], and 6 case reports [20,23,31,32,33,34]. In two-thirds (66.6%) of the RCT included, the authors reported beneficial effects immediately after physiotherapy and in the long-term in pain, function, muscle strength, sensory thresholds, and nerve conduction [13,30]. One RCT reported no significant treatment effect on pain scores, muscle strength, function, and health status [29]. In all case studies, the authors emphasised the positive effects of different physiotherapy procedures concerning the reduction in pain symptoms, improved function, increased pinch-grip strength, and a reduction in symptoms in provocation tests (Tinel’s sign, elbow flexion test), both immediately after treatment [20,21,22,23,31,32,33] and in the long-term follow-up [21,22,23,32,33,34]. 

Although 91% of the included papers reported beneficial effects of physiotherapy [13,20,21,22,23,31,32,33,34], the results obtained have to be viewed with caution. This is due to the fact that 73% of current publications were case studies [20,21,22,23,31,32,33,34] with small numbers of subjects (24 in total), very heterogeneous research methodology and physiotherapy programs, and a high risk of bias. Furthermore, 3 RCTs [13,29,30] examined 163 people, but even in this case, due to methodological differences, it is difficult to draw firm conclusions about the effectiveness of physiotherapy treatment in CuTS, particularly when one RCT did not observe any significant effect [29].

It should be pointed out that the method used for the diagnosis of CuTS was problematic in most of the papers included in the review. Although a nerve conduction study (NCS) was performed in 54% of papers [13,22,29,30,32,33], the result was negative in one paper, despite the symptoms that could indicate CuTS [32]. In the other papers, the diagnosis was based on clinical and orthopaedic examinations [20,21,23,31,34]. As NCS-based diagnosis of peripheral neuropathies is the gold standard [35] in some compression neuropathies (as is the case in CTS), it appears that it should also apply to other peripheral neuropathies. According to the accepted study protocol, it was decided to include the analysis papers in which patients were diagnosed as CuTS despite the absence of a nerve conduction study but only based on history and clinical and orthopaedic examinations. Therefore, the results of some studies and their findings must be viewed with caution. It seems that electrodiagnostic tests are necessary not only to make an accurate diagnosis of CuTS but also to assess the severity of the condition and objectively evaluate improvements after therapy [36]. Although some authors have demonstrated the high sensitivity and specificity of clinical tests in the diagnosis of CuTS [37,38], there is no agreement [39]. It is important to note that, regardless of the differences in the assessment of the various clinical tests, this will be a much more subjective assessment than it is in the case of NCS. Furthermore, NCS provides specific information such as conduction velocity in nerve fibres, latency, and amplitude, information that cannot be obtained from a functional test. However, it should also be noted that some authors indicated that symptoms often precede nerve conduction disturbances [40,41], especially in the early stages of peripheral neuropathy. In these cases, for obvious reasons, the diagnosis must be based only on clinical symptoms and orthopaedic examination, which can also explain the failure to use NCS study in some papers to some extent. It would be unethical to omit clinical cases or delay therapeutic management in patients who have subjective symptoms but do not have a disrupted NCS. On the other hand, the lack of a certain diagnosis does not allow the use of therapy aimed at the cause but only at reducing the patient’s symptoms.

Significant differences in the offered therapeutic programs are another major problem in assessing the effectiveness of physiotherapy approaches in CuTS based on the papers included in this review. In fact, it is difficult to find any similarities in the therapeutic programs used. In 64% of the papers, physiotherapy was based just on one form of therapy [13,20,22,23,29,30,33]. However, in each of them, this was a different type of therapy. The remaining 36% of papers [21,31,32,34] used a total of multiple therapeutic measures, not only targeting the elbow region but also the cervical spine and thorax [31,32]. Neurodynamic techniques were used in 36% of the papers [13,21,32,34] but with different methodologies and often as a component of other treatments. In 18% of the papers, neurodynamic techniques were used as a self-therapy program, which may make it even more difficult to control the correctness of their performance and regular use. 

A major weakness of all qualified papers is their low methodological quality, which was confirmed by the RoB results. Regardless of study type (case series, case reports, or RCTs), low RoB was not reported in any of the papers included. Therefore, this review indicated a complete lack of high-quality studies evaluating the effectiveness of physiotherapy in the conservative treatment of CuTS. Taking into account the prevalence of CuTS and the strong evidence confirming the effectiveness of physiotherapy in other peripheral neuropathies [15,16,17,18,19], it is difficult to indicate the reason for such low interest in CuTS conservative treatment.

### 4.1. Limitations

This systematic review has some potential limitations. The number of papers included in the review was small, and only three RCTs were identified. The sample size was small, with only 186 subjects. Some of the papers lacked clinical information about the patients’ condition and severity of CuTS, which may have also affected the obtained results. The lack of an NCS study conducted not only to make an accurate diagnosis but also to assess the effects achieved after the therapeutic cycle is another limitation of many of the included papers and, thus, of the review itself. Only papers published in English were included in this review, which may have resulted in the omission of some studies evaluating the effectiveness of physiotherapy in the conservative treatment of CuTS. Further, the papers included in the review were characterised by different therapeutic programs, which made it difficult to assess the therapeutic effects.

### 4.2. Implications for Future Research

The results of this review allow for several recommendations for future research. First of all, there is a need for more well-designed RCT studies with two or more representative groups of subjects. Due to the prevalence of CuTS, case series and case reports have no substantive justification. It is also important to perform the diagnosis of CuTS based on NCS and ultrasound imaging, while clinical symptoms and other tests should be considered as supplementary information about the study population. The diagnosis of CuTS should be supplemented with the determination of the CuTS stage, which is also important in the assessment of the effectiveness of the offered therapy. Since most of the papers included in the review had a beneficial therapeutic effect, randomised controlled trials should be conducted based on similar therapeutic programs such as manual therapy including neurodynamic techniques, ultrasound therapy, low-level laser therapy, percutaneous electrical stimulation, or dry needling. Future studies should be multi-central and (if possible) blinded or even double-blinded.

## 5. Conclusions

Although physiotherapy could have the potential to demonstrate a positive effect in the treatment of CuTS, most published studies to date are of questionable methodological quality. Thus, at this stage of knowledge, there is no possibility to recommend the best method, duration, and interval of physiotherapy in the clinical practice of people with CuTS. More high-quality studies are required.

## Figures and Tables

**Figure 1 jcm-11-04247-f001:**
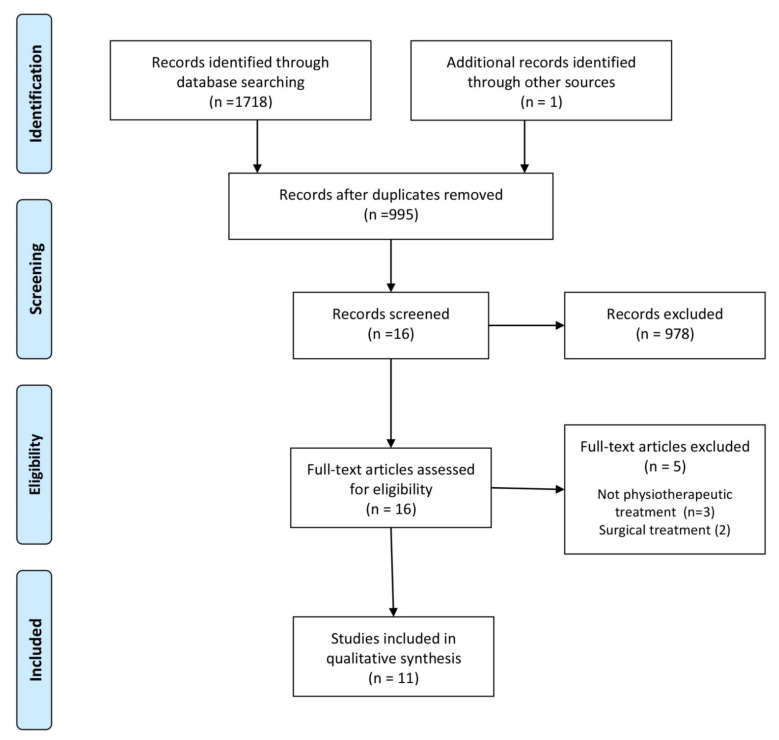
Flowchart.

**Table 1 jcm-11-04247-t001:** Characteristics of included randomised clinical trials (RTCs).

Study	Participants	Outcome Measures	Control/ Comparison	Intervention	Results
Svernlov et al. [13]	*n* = 70 Sex: 39 female; 31 male Age: 17–72	Measurement at baseline and 6-month follow-up; Activities scale (COPM); Grip strength (JAMAR dynamometer); Pain (VAS); Neurophysiological examination (NCS, electromyography).	Three groups	Group A—elbow orthosis (3-month treatment); Group B—nerve gliding exercises (3-month treatment); Group C—information (exercise modification).	There was a significant improvement in each group after the applied therapy. There were no between-group differences.
Badur et al. [29]	*n* = 61 Sex: 32 female; 29 male Age: 16–79	Measurement at baseline, after treatment, and 1- and 3-month follow-up; Pain (VAS); Upper-extremity disability and symptoms (DASH); Overall health (SF-36 questionnaire); Grip strength (dynamometer).	Two groups	Group 1—continuous shortwave diathermy (10 treatments); Group 2—placebo shortwave diathermy (10 treatments).	There were no changes in the assessed parameters in both groups after the applied therapy.
Ozkan et al. [30]	*n* = 32 Sex: 16 female; 16 male Age: mean = 43.5	Measurement at baseline, after treatment, and 1- and 3-month follow-up; Pain (VAS); Grip strength (dynamometer); Sensory threshold (Semmes–Weinstein monofilament test; Neurophysiological examination (NCS); patient satisfaction scale.	Two groups	Group 1—low-level laser therapy (10 treatments); Group 2—ultrasound therapy (10 treatments).	There was a significant improvement in both groups after the applied therapy. There were no between-group.

COPM—Canadian Occupational Performance Measure; VAS—visual analogue scale; NCS—nerve conduction study; DASH—Disabilities of the Arm, Shoulder, and Hand Outcome Measure.

**Table 2 jcm-11-04247-t002:** Characteristics of included case studies design.

Study	Participants	Outcome Measures	Control/ Comparison	Intervention	Results
Kearns and Wang [20]	*n* = 1 Sex: female Age: 45	Measured at baseline and 4 weeks post-treatment; Tissue tension (STTT); range of motion (goniometer); symptom provocation (ULTT, elbow flexion test); structural dysfunction (PAM); pain (NPRS).	No	Thrust manipulation (humeroulnar joint 2 treatments, radiocarpal joint 1 treatment).	After 3 treatment sessions, pain and paraesthesia were resolved; all other tests were negative.
Oskay et al. [21]	*n* = 7 Sex: not specified Age range: 35–70	Measured at baseline, after treatment, and 12-month follow-up; Symptom provocation (elbow flexion test); palmar gripping and grasping (pinchmeter and grip dynamometer); pain (VAS); loss of sensation (Semmes–Weinstein monofilaments); upper-extremity disability and symptoms (DASH).	No	Cold application; pulsed ultrasound (10 treatments); nerve mobilisation techniques (10 treatments); strengthening exercises; postural adaptations; education; ergonomic modifications.	Pain, Tinel’s sign, and Disability of Arm, Shoulder, and Hand Index scores were decreased; grip and pinch strength increased during the observation period.
Shen et al. [22]	*n* = 7 Sex: not specified Age: 35–71	Measured at baseline and 4-, 8-, and 12-week follow-up; Severity of paraesthesia/dysaesthesia (VAS); upper-extremity disability and symptoms (DASH).	No	Extracorporeal shock wave therapy (3 treatments).	The VAS and Quick DASH scores demonstrated improvements at all follow-up time points in all treated elbows.
Anandkumar and Manivasagam [23]	*n* = 3 Sex: 2 male; 1 female Age: 35,45,50	Measured at baseline, beginning of each treatment session, and at 6-month follow-up; Pain (NPRS); function limitation (PSFS); pain-free grip strength (JAMAR dynamometer); self-reported outcome measure (GROC).	No	Dry needling (4 treatments, twice a week).	All patients achieved complete pain reduction and full recovery of function; the strength of the pain-free grip also improved; all self-reported significant improvement after therapy; the effect lasted 6 months after the therapy.
Illes and Johnson [31]	*n* = 1 Sex: female Age: 41	Measured at baseline and after treatment; Severity of numbness (VAS); Symptom provocation (EAST, Tinel sign); Grip strength (Blood pressure cuff);	No	Chiropractic manipulative therapy; myofascial therapy (11 treatments); elastic therapeutic taping (no number specified); home exercises (8 treatments).	After 11 treatment sessions, symptoms resolved completely.
Coppieters et al. [32]	*n* = 1 Sex: female Age: 17	Measured at baseline, beginning of each treatment session, and at 6- and 10-month follow-up; Pain (VAS); range of motion (goniometer); clinical tests (neural provocation test, elbow flexion test, Tinel’s sign); functional status (NPQ).	No	Neurodynamic mobilisation (5 treatments); Elbow mobilisation (4 treatments); home exercises—active ulnar nerve-sliding (5 treatments); high-velocity distraction/rotation thrust (3 treatments); education (1 instruction).	After the applied therapy, in each of the tests used, the symptoms were eliminated; The effect lasted 10 months after the therapy.
Kwak et al. [33]	*n* = 2 Sex: male Age: 39 and 40	Measured at baseline, after treatment, and at 1, 2, 3, and 6 months post-treatment; Pain (NPRS); NCS; elbow imaging (MRI).	No	PRF (1 treatment).	After 1 treatment session, the pain was completely relieved. At the 1-, 2-, 3-, and 6-month follow-up assessments after the procedure, the previously reported pain had not recurred.
Fernández-de-Las-Peñas al. [34]	*n* = 1 Sex: male Age: 48	Measured at baseline and at 1, 3, 6, and 12 months post-treatment; Upper-extremity disability and symptoms (DASH); neuropathic pain (S-LANSS); self-reported outcome measure (GROC).	No	PENS of the ulnar nerve (3 treatments); self-neural glides as a home program (2–3 weeks).	After three treatment sessions, there was an elimination of pain and symptoms and an improvement in functional status; the effect lasted 12 months after the therapy.

NPRS—Numeric Pain Rating Scale; PSFS—Patient Specific Functional Scale; GROC—global rating of change; VAS—visual analogue scale; NPQ—Northwick–Park Questionnaire; DASH—Disabilities of the Arm, Shoulder, and Hand Outcome Measure; S-LANSS—Leeds Assessment of Neuropathic Symptoms and Signs; PENS—ultrasound-guided percutaneous electrical stimulation; STTT—selective tissue tension test; ULTT—upper-limb tension test; PAM—passive accessory movement; PRF—pulsed radiofrequency; NCS—nerve conduction study; MRI—magnetic resonance Imaging.

**Table 3 jcm-11-04247-t003:** Risk of Bias in randomised clinical trials (RTCs).

Study	Domain 1 Risk of Bias Arising from the Randomisation Process	Domain 2 Risk of Bias Due to Deviations from the Intended Interventions	Domain 3 Missing Outcome Data	Domain 4 Risk of Bias in Measurement of the Outcome	Domain 5 Risk of Bias in Selection of the Reported Result	Overall Risk of Bias
Svernlov et al. [13]	Low	Some concerns	Low	High	Low	High
Badur et al. [29]	Low	Some concerns	Low	Low	Low	Some concerns
Ozkan et al. [30]	Low	Some concerns	Low	Low	Low	Some concerns

**Table 4 jcm-11-04247-t004:** Critical appraisal checklist for case studies design.

Critical Appraisal Checklist	Kearns and Wang [20]	Oskay et al. [21]	Shen et al. [22]	Anandkumar and Manivasagam [23]	Illes and Johnson [31]	Coppieters et al. [32]	Kwak et al. [33]	Fernández-de-Las-Peñas et al. [34]
1. Were the patient’s demographic characteristics clearly described?	Yes	Unclear	Yes	Yes	Yes	Yes	Yes	Yes
2. Was the patient’s history clearly described and presented as a timeline?	Yes	Yes	Yes	Yes	Yes	Yes	Yes	Yes
3. Was the current clinical condition of the patient on presentation clearly described?	Yes	Yes	Yes	Yes	Yes	Yes	Yes	Yes
4. Were diagnostic tests or assessment methods and the results clearly described?	Unclear	Yes	Yes	Unclear	Unclear	Unclear	Yes	Unclear
5. Was the intervention(s) or treatment procedure(s) clearly described?	Yes	Yes	Yes	Yes	Yes	Yes	Yes	Yes
6. Was the post-intervention clinical condition clearly described?	Yes	Yes	Yes	Yes	Yes	Yes	Yes	Yes
7. Were adverse events (harms) or unanticipated events identified and described?	No	Unclear	Unclear	Unclear	No	Unclear	Unclear	No
8. Does the case report provide takeaway lessons?	No	Yes	Yes	Yes	No	No	Yes	Unclear

**Table 5 jcm-11-04247-t005:** Risk of bias in case studies design.

Study	Bias Due to Confounding	Bias in Selection of Participants in the Study	Bias in Classification of Interventions	Bias Due to Deviations from Intended Interventions	Bias Due to Missing Data	Bias in Measurement of Outcomes	Bias in Selection of the Reported Result	Overall Bias
Kearns and Wang [20]	Serious	Moderate	Serious	Serious	Moderate	Serious	Moderate	Serious
Oskay et al. [21]	Serious	Moderate	Serious	Serious	Moderate	Serious	Moderate	Serious
Shen et al. [22]	Moderate	Low	Moderate	Moderate	Low	Moderate	Moderate	Moderate
Anandkumar and Manivasagam [23]	Moderate	Low	Moderate	Moderate	Low	Moderate	Moderate	Moderate
Illes and Johnson [31]	Serious	Moderate	Serious	Serious	Moderate	Serious	Moderate	Serious
Coppieters et al. [32]	Serious	Moderate	Serious	Serious	Moderate	Serious	Moderate	Serious
Kwak et al. [33]	Moderate	Low	Moderate	Moderate	Low	Moderate	Moderate	Moderate
Fernández-de-Las-Peñas et al. [34]	Serious	Moderate	Serious	Serious	Moderate	Serious	Moderate	Serious

## Data Availability

Data are available upon request.

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
