# Peer review of "The Effects of Physiotherapy in the Treatment of Cubital Tunnel Syndrome: A Systematic Review"

_jcm, 2022, doi:10.3390/jcm11144247_

Round 1

Reviewer 1 Report

The authors conducted a systematic review for conservative treatment of cubital tunnel syndrome. They included three RCT (level 1 study) studies and eight case series/report (level 4 or 5) studies.

#1 The authors did not include level 2 or 3 studies while a similar study (Conservative treatment of cubital tunnel syndrome: a systematic review, Sahil Kooner et al. 2019.) included 4 level II and 3 level III studies. Please explain.

#2 More than three cases are usually considered a case series. Thus, this manuscript included six case reports and two case series. Although case reports can be included in a systematic review, I think the effect could be little. The authors should clarify this.

#3. Thus, this manuscript included three systematic reviews of 163 patients, two case series of 14 patients, and six case reports of 8 patients. This is not a much-advanced manuscript from Sahil Kooner’s systematic review. The authors should include level II or III studies and should classify case series and reports.

Author Response

Reviewer 1

The authors conducted a systematic review for conservative treatment of cubital tunnel syndrome. They included three RCT (level 1 study) studies and eight case series/report (level 4 or 5) studies.

Comment 1: The authors did not include level 2 or 3 studies while a similar study (Conservative treatment of cubital tunnel syndrome: a systematic review, Sahil Kooner et al. 2019.) included 4 level II and 3 level III studies. Please explain.

Response: We would like to thank you for the effort to review our manuscript. We have read the article you have mentioned.  We have also cited this work (position 5) in our systematic review. However, Kooner et al. (2019) have performed their systematic review taking into account conservative treatment of cubital tunnel syndrome (CuTS), while we were only focused on physiotherapy. There was the reason why some of studies included in Kooner`s review were not included in our review. Initially, we have included all type of studies - from level I (high-quality RCT) to levels IV (cases series or case report) – see method section. In fact, the review by Sahil Kooner et al. 2019 included several papers on local steroid injections (levels II and III studies) which were out of the scope of our review.  

Comment 2: More than three cases are usually considered a case series. Thus, this manuscript included six case reports and two case series. Although case reports can be included in a systematic review, I think the effect could be little. The authors should clarify this.

Response: Thank you for your valuable comment. Agree. We have corrected our manuscript to specify case series and reports.   

Comment 3: Thus, this manuscript included three systematic reviews of 163 patients, two case series of 14 patients, and six case reports of 8 patients. This is not a much-advanced manuscript from Sahil Kooner’s systematic review.

Response: We do not agree with this opinion. Our systematic review for the first time assess the effectiveness of physiotherapy treatment for CuTS. We have included all type of studies and all studies were evaluated in relation to risk of bias.  The main differences between our and Sahil Kooner’s systematic review are as follow:

  1. Kooner et al. did not take into account methodological quality assessment of included studies. In our review, all articles were evaluated in relation to risk of bias by two independent researchers.
  2. Kooner`s search was finished in September 2017. In our review the search was finished in March 2022. One of the included RCT was not included in Kooner`s review as was published in 2020.
  3. Kooner et al. included in their publication only three out of the eleven articles which were included in our review. This means that 8 articles were not even mentioned in Kooner et al. study. From clinical perspective, the Kooner et al. review omitted shortwave diathermy, extracorporeal shock wave therapy, thrust manipulation, dry needling, chiropractic therapy, percutaneous electrical stimulation, pulsed radiofrequency. All of these modalities may be clinically relevant and were included in our systematic review.
  4. We agree that Kooner`s review present more form of treatment (all conservative), but our aim was to deeply focus on all physiotherapy interventions. Therefore, our review is a complete overview of physiotherapy in CuTS. Systematic review by Kooner et al. cannot be treated in this way and seems not be as useful as our review for physical therapist.

Reviewer 2 Report

In this review of the physiotherapy that is used in the treatment of cubital tunnel syndrome (CuTS), the authors with a limited number of studies did not reach any conclusion about physiotherapy effectiveness, only recommend more homogeneous RTC with more patients.

 It is remarkable that the need for objective measures like NCS-based diagnosis has only been reported by few studies.

 The paper is well written but with limited conclusions. For that, I recommend in point 4.2 that included the most accurate possible general scheme of the RCT in which we could extract the conclusions of physiotherapy effectiveness

 Minor Issues

 1) Appendix I with the search parameters is not included.

 2) In Table 5, the last row is not readable.

Author Response

Comment: In this review of the physiotherapy that is used in the treatment of cubital tunnel syndrome (CuTS), the authors with a limited number of studies did not reach any conclusion about physiotherapy effectiveness, only recommend more homogeneous RTC with more patients.

It is remarkable that the need for objective measures like NCS-based diagnosis has only been reported by few studies. The paper is well written but with limited conclusions.  

Response: Thank you for words of appreciation. The conclusions are limited as we were unable to perform proper meta-analysis.

Comment: For that, I recommend in point 4.2 that included the most accurate possible general scheme of the RCT in which we could extract the conclusions of physiotherapy effectiveness

Response: We had added new information in section 4.2 following reviewer's recommendation.

We have added the following sentence:

Future studies should be multi-centrally and (if possible) blinded or even double-blinded.

Minor Issues

 1) Appendix I with the search parameters is not included.

 Response: Appendix I was included in the manuscript as another file. I`m sorry that you were unable to evaluate it.  I hope that in the revised version this file will be accessible.

2) In Table 5, the last row is not readable.

Response: Table 5 has been corrected.

Reviewer 3 Report

I commend the interesting systemic review carried out, the results of which I hope will be useful in enabling further technical comparisons among practitioners to explore technical aspects that can certainly be helpful to those involved in the treatment of this condition. I believe that the information provided is to be considered entirely sufficient, appropriate and well articulated.

the topic brought forward by this paper is very interesting:

- the work is well written, the structure solid and in line with current scientific knowledge.

- the study design is rational and consistent.

- the conclusions presented are significant and the statistical sample, on which the findings are based, is adequate.

- the scientific value of this work is relevant.

Author Response

I commend the interesting systemic review carried out, the results of which I hope will be useful in enabling further technical comparisons among practitioners to explore technical aspects that can certainly be helpful to those involved in the treatment of this condition. I believe that the information provided is to be considered entirely sufficient, appropriate and well-articulated.

 the topic brought forward by this paper is very interesting:

- the work is well written, the structure solid and in line with current scientific knowledge.

- the study design is rational and consistent.

- the conclusions presented are significant and the statistical sample, on which the findings are based, is adequate.

- the scientific value of this work is relevant.

Response: Thank you for words of appreciation.

Round 2

Reviewer 1 Report

All comments were amended properly.